# The Effect of Capsaicin on Growth Performance, Antioxidant Capacity, Immunity and Gut Micro-Organisms of Calves

**DOI:** 10.3390/ani13142309

**Published:** 2023-07-14

**Authors:** Minqiang Su, Yuanhang She, Ming Deng, Yongqing Guo, Yaokun Li, Guangbin Liu, Hui Zhang, Baoli Sun, Dewu Liu

**Affiliations:** 1College of Animal Science, South China Agricultural University, Guangzhou 510642, China; smq2054113415@163.com (M.S.); syh15521170780@163.com (Y.S.); dengming@scau.edu.cn (M.D.); yongqing@scau.edu.cn (Y.G.); ykli@scau.edu.cn (Y.L.); gbliu@scau.edu.cn (G.L.); 2State Key Laboratory of Swine and Poultry Breeding Industry, South China Agricultural University, Guangzhou 510642, China; 3Collaborative Innovation Center for Healthy Sheep Breeding and Zoonoses Prevention and Control, Shihezi University, Shihezi 832000, China; prof.zhang@foxmail.com

**Keywords:** capsaicin, calves, antioxidant capacity, immunity, gut microbiota

## Abstract

**Simple Summary:**

Pre-weaned calves have immature immune systems, lack immunocompetence, and often suffer from stress and disease. Diarrhea is a common disease in nursing calves. Antibacterials are widely used in calf production as a common treatment for diarrhea. However, the extensive use of antibacterials can lead to the development of drug-resistant pathogens and endanger public health and safety. Capsaicin is a natural plant extract with antibacterial, anti-inflammatory and antioxidant bioactivities. In this study, we investigated the effects of capsaicin on growth performance, fecal scores, fecal fermentation parameters, antioxidant and immune capacity and the gut microbiota of nursing calves. The results showed that the addition of capsaicin had no effect on calf growth performance and fecal fermentation parameters, improved the antioxidant and immune abilities of calves and also improved the gut microbial environment, which was beneficial for the healthy growth of calves.

**Abstract:**

Capsaicin is the active ingredient of the red pepper plant of the genus *Capsicum*. The aim of this study was to investigate the effects of different doses of capsaicin on growth performance, antioxidant capacity, immunity, fecal fermentation parameters and gut microbial composition in nursing calves. Twenty-four newborn Holstein calves were randomly assigned to three treatment groups, which each consisted of eight calves. The milk replacer was supplemented with 0, 0.15 or 0.3 mL/d of capsaicin in each of the three treatment groups. During the 4-week experiment, intake was recorded daily, body weight and body size parameters were measured at the beginning and end of the trial and serum samples and rectal fecal samples were collected at the end of the trial to determine serum parameters, fecal fermentation parameters and fecal microbiome compartments. The results showed that both doses of capsaicin had no negative effect on the growth performance or the fecal fermentation parameters of calves, and the higher dose (0.3 mL/d) of capsaicin significantly improved the antioxidant capacity and immunity of calves. The calves in the high-dose capsaicin-treated group had lower fecal scores than those recorded in the control group. High doses of capsaicin increased glutathione antioxidant enzyme, superoxide dismutase, immunoglobulin A, immunoglobulin G, immunoglobulin M and interleukin-10 levels and decreased malondialdehyde and bound bead protein levels. In addition, capsaicin regulated the gut microbiota, reducing the abundance of diarrhea-associated bacteria, such as *Eggerthella*, *Streptococcus*, *Enterococcus* and *Enterobacteriaceae*, in the gut of calves in the treated group. Therefore, high doses of capsaicin can improve the antioxidant and immune capacity of calves without affecting growth performance, as well as improve the gut microbiological environment, which enables the healthy growth of calves.

## 1. Introduction

Due to the immature development of the immune system and the immature immune response of pre-weaned calves, they are prone to stress and disease when faced with critical windows. In calves, stress and the development of the immune system can affect their overall health and performance in later life. Diarrhea is one of the major causes of morbidity and death in nursing calves [1,2]. Most cases of diarrhea in calves occur in calves that are less than one month old [3]. The calves’ diarrhea can be caused by both infectious and non-infectious factors, mainly including infectious diarrheal pathogens, feeding management and calves’ immune status [4]. Antibacterial are widely used in farm production as a common treatment for diarrhea. However, the development of resistant pathogens as a result of heavy antibacterial use can be a serious health risk for livestock and humans [5]. In addition, long-term use of antibacterial may disrupt the gut microbial ecology of calves [6]. The development of plant extracts such as those used as alternatives to antibacterial treatments could significantly reduce the use of antibiotics in the breeding industry.

Capsicum annuum is a herb of the genus *Capsicum* in the family Solanaceae, and it is used for medicinal purposes. Capsaicin (CAP) has been reported as the active component of red pepper, which has biological activities such as antibacterial, anti-inflammatory and antioxidant properties, as well as stimulating small intestine development [7]. The addition of CAP to the diet of lactating cows increased DMI [8]. Other studies have shown that feeding CAP to cows has no effect on DMI or nutrient digestibility [9]. CAP antibacterial activity has been shown to be associated with *Salmonella typhimurium*, *Pseudomonas aeruginosa*, *Escherichia coli*, *Staphylococcus aureus*, *Bacillus subtilis*, *Streptococcus pyogenes*, *Helicobacter pylori*, *Staphylococcus pyogenes* and *Aspergillus niger* [10,11,12]. Experimental animal studies revealed that CAP reduced the incidence and severity of foot inflammation and delayed the onset of adjuvant-induced foot inflammation in rats [13,14]. In addition, CAP inhibited the peroxidation reaction, and its inhibitory effect was stronger than that of a-tocopherol, which is a well-known antioxidant [15]. Therefore, the aim of this study was to evaluate the effects of adding different doses of CAP on growth performance, fecal scores, fecal fermentation parameters, antioxidant and immune capacity and gut microbial composition of nursing calves. The results of this study provide a theoretical basis for the application of CAP in nursing calves.

## 2. Materials and Methods

This study was conducted from November 2022 to December 2022 in a commercial dairy farm in Qingyuan City, Guangdong Province, China. The average temperature was 17.1 °C (5.8–31.4 °C), while the average humidity was 64.2% (14.8–85.7%), during the time period in which the experiment was conducted.

### 2.1. Experimental Design and Treatments

Twenty-four newborn Chinese Holstein female calves were randomly selected to participate in this feeding experiment (BW = 36.67 ± 3.44 kg). The test calves were separated from their dams immediately after birth, weighed and fed a volume of qualified colostrum equivalent to 10% of calf body weight. The colostrum was thawed in advance using a colostrum pasteurization machine (>22% Brix). Blood samples were collected through the jugular vein 48 h after colostrum feeding, and total serum protein concentrations were determined using a portable refractometer, with successful passive immunization determined based on exceeding 5.5 g/dL. In this study, all test calves exceeded the threshold serum total protein concentration, with an average rate of 6.7 ± 0.7 g/dL. Eight groups of three calves were divided according to birth order and similar birth weight, and one calf in each group was randomly selected and assigned to one of three treatment groups: group C (no capsaicin addition, n = 8), group L (1% capsaicin 0.15 mL/day/head, n = 8) or group H (1% capsaicin 0.3 mL/day/head, n = 8). The capsaicin used was 1% water-soluble capsaicin, and the other contents were emulsifier and water, which were produced by the Leader Biotechnology Company, Guangzhou, China.

### 2.2. Housing, Management and Dietary Treatments

All test calves started the formal experiment at 5 days of age, and the experiment lasted for one month until reaching 35 ± 2.04 days of age. All calves were housed in individual pens (1.6 m long, 0.9 m wide and 1.5 m high), with partitions placed between pens to prevent calves from touching each other. In front of each pen, there were water and feed buckets for calves to drink and feed freely, and buckets were changed every morning. The bedding was changed every two days, with straw used as bedding. Milk replacer (MR) was prepared in a 1:6 ratio of milk powder and water (42 °C) and bottle-fed at 07:00 and 16:00 daily at 10 L/d (5 L/meal). Before each feeding, 0.15 mL/d (0.075 mL/meal) or 0.3 mL/d (0.15 mL/meal) of capsaicin was added to 1.5 L of MR in the corresponding treatment group and mixed well before feeding. We ensured that all MR-containing capsaicin was consumed before feeding the remaining MR.

### 2.3. Feed Analysis

Feed samples, including roughage, calf starter, leftovers and milk replacer, were collected weekly and mixed during each period. Dry matter (DM) was measured via drying at 100 °C in a forced-air oven for 24 h, and the crude ash content (Ash) was measured via furnace incineration at 550 °C for 4 h [16]. Dried samples were crushed using a grinder and passed through a 40-mesh screen. The samples present after grinding through a sieve were used to determine crude protein (CP) using Kjeldahl nitrogen [17], ether extract (EE) via Soxhlet extraction [16], acidic detergent fiber (ADF) and neutral detergent fiber (NDF) via a fiber meter using anhydrous sodium sulfate and heat-stabilized amylase, according to the method published by Van Soest et al. [18]. Non-fiber carbohydrates (NFC) of feed samples were estimated according to Mertens’ method [19]: NFC (%) = 100% − (NDF% + CP% + fat% + ash%). The results of the feed nutrient determination are provided in Table 1.

### 2.4. Determination of Growth Performance

Daily detailed records of calf starter feeds and leftovers were kept, and MR and dry matter of calf starter were used for total DMI and calculation of the feed efficiency. Calves were weighed on the first (5 d) and last (35 ± 2.04 d) days of the formal trial, and all calves were weighed using electronic scales before morning feeding. Calves were also measured at all tow periods for body measurements, including body height, body length, body slant length and chest.

### 2.5. Health Exams

All calves were scored daily for feces by the same trained veterinarian. The scoring system followed the method of Renaud et al. [20], consisting of 4 levels: 0 = normal (firm but not hard, original form slightly distorted after falling to the ground and settling); 1 = soft (did not retain form, piled up but spread slightly); 2 = runny (spread easily); and 3 = watery (liquid consistency, splashing). Calf diarrhea is considered present if a fecal score of ≥2 occurs for more than 2 consecutive days. Oral rehydration solution (Guangdong Wens Dahuanong Biotechnology Co., Ltd., Yunfu, China) was provided daily to the diarrheic calves, consisting of 50 g of glucose, 20 g of sodium bicarbonate and 10 g of sodium chloride dissolved in 2 L of warm water. The administration of oral rehydration solution was discontinued when the fecal score reached ≤ 1.

### 2.6. Determination of Serum Parameters

Approximately 10 mL of jugular vein blood samples were collected 2 h before morning feeding on the first and last days of the trial using a separator gel pro-coagulation tube. Blood samples were centrifuged for 15 min at 3500 r/min in a low-speed centrifuge, and the serum was separated, collected in 1.8 mL centrifuge tubes and stored at −20 °C for serum biochemical and metabolite measurements. Glucose (Glu), glutamic aminotransferase (ALT), glutamic oxalacetic transaminase (AST), alkaline phosphatase (ALP), total protein (TP), albumin (ALB), urea nitrogen (BUN) and creatinine (CR) were measured using a ZeCheng fully automated biochemical instrument (CLS880, Weifang, China). Immunoglobulin A (IgA), Immunoglobulin G (IgG) and Immunoglobulin M (IgM) were determined via ELISA according to the kit’s instructions (BGI Jiuzhou Taikang Biotechnology Co., Ltd., Beijing, China). Interleukin-1β (IL-1β), interleukin-6 (IL-6), interleukin-10 (IL-10), tumor necrosis factor-α (TNF-α), amyloid A (SAA) and binding bead protein (HP) were determined via ELISA according to the kit’s instructions (Shanghai Zhangshi Biotechnology Co., Ltd., Shanghai, China). Total antioxidant capacity (T-AOC), malondialdehyde (MDA), glutathione dismutase (GSH-PX) and superoxide dismutase (SOD) were determined via the colorimetric method according to the kit’s instructions (Nanjing Jiancheng Institute of Biological Engineering, Nanjing, China). β-hydroxybutyric acid (BHBA) was determined via the dehydrogenase method according to the kit’s instructions (Shanghai Zhangshi Biotechnology Co., Ltd., Shanghai, China).

### 2.7. Fecal Sample Collection

On the last day of the experiment, 10 g of rectal feces was taken via the intestinal invasion method 2 h after morning feeding. Next, 2 g of feces was stored in 1.8 mL sterile polypropylene tubes and snap frozen in liquid nitrogen for fecal microbiome analysis. The remaining feces were stored in centrifuge tubes at −20 °C for short-chain fatty acid (SCFA) determination.

### 2.8. Determination of SCFAs in Rectal Feces

For sample pre-treatment, the fecal sample of the rectum was thawed at room temperature, weighed to 1 g, mixed well with 6 mL of ultrapure water and left overnight at 4 °C. Centrifuging occurred at 4000 r/min for 10 min at 4 °C, 1 mL of supernatant was added to a 1.5-milliliter centrifuge tube, and 0.2 mL of metaphosphoric acid solution that contained the internal standard 2-ethylbutyric acid (we weighed 25 g of metaphosphoric acid and 0.217 mL of 2-ethylbutyric acid using a 100 mL volumetric flask, and we fixed the volume with ultrapure water to 100 mL to configure a 25% (*w*/*v*) solution of metaphosphoric acid that contained 2 g/L of 2-ethylbutyric acid) was added. After mixing, the solution was placed in an ice-water bath for 30 min and centrifuged at 10,000 r/min for 10 min at 4 °C. After centrifugation, 1 mL of supernatant was collected and filtered through a 0.22-micrometer filter membrane into a super-sampling bottle.

For standard solution pre-treatment, the standard solution consisting of 330 μL of acetic acid, 400 μL of propionic acid, 30 μL of isobutyric acid, 160 μL of butyric acid, 40 μL of isovaleric acid and 50 μL of valeric acid was added to a 100-milliliter volumetric flask. The flask was then filled with ultrapure water up to the 100 mL mark, resulting in the preparation of a mixed standard stock solution. Next, we added 0.2, 0.15, 0.1, 0.05 and 0.025 mL of the mixed standard stock solution into 1.5 mL centrifuge tubes. We then added 0, 0.05, 0.1, 0.15 and 0.175 mL of ultrapure water to the corresponding centrifuge tubes to create five different gradients of standard solutions. We added 0.2 mL of metaphosphoric acid solution containing the internal standard 2-ethylbutyric acid to each centrifuge tube. We then mixed well and filtered through a 0.22-micrometer filter membrane into a super-sampling bottle.

Analysis was performed using an Agilent-7890 meteorological chromatograph. The detection procedure used was as follows: A DB-FFAP column of 60 m × 0.25 mm × 0.50 μM was used. The carrier gas was high-purity nitrogen (99.999%) at a flow rate of 0.8 mL/min, and the auxiliary gas was high-purity hydrogen (99.999%). The detector FID temperature was 250 °C, the inlet temperature was 220 °C, the split ratio was 40:1 and the injection volume was 1.5 μL. The ramp-up procedure was 20 °C/min at an initial temperature of 60 °C rising to 120 °C over 3 min, before rising to 180 °C over 3 min.

### 2.9. Fecal Microbiome Analysis

Fecal DNA was extracted using the E.Z.N.A. Soil DNA Extraction Kit (Omega Bio-Tek, Norcross, GA, USA). The V3-V4 region of the bacterial 16S rRNA gene was amplified using forward primer 338F (5′-ACTCCTACGGGAGGCAGCA-3′) and reverse primer 806R (5′-GGACTACHVGGGTWTCTAAT-3′). The PCR system contained 5 μL of buffer (5×), 0.25 μL of Fast pfu DNA polymerase (5 U/μL), 2 μL of dNTPs (2.5 mM), 1 μL of forward and reverse primers (10 μM), 1 μL of DNA template and 14.75 μL of ddH_2_O. PCR conditions included pre-denaturation at 98 °C for 5 min, followed by 30 cycles (denaturation at 98 °C for 30 s, annealing at 53 °C for 30 s and extension at 72 °C for 45 s). The final extension was at 72 °C for 5 min. The PCR products were recovered via electrophoretic detection, followed by magnetic bead purification. The recovered PCR products were quantified via fluorescence using the Quant-iT PicoGreen dsDNA Assay Kit (Thermo Fisher Scientific, Waltham, MA, USA) and a Microplate reader (BioTek, FLx800, Winooski, VT, USA). Based on the fluorescence quantification results, each sample was mixed in the appropriate ratio according to the sequencing volume requirement of each sample, and sequencing libraries were constructed using the NovaSeq 6000 SP kit (500 cycles). Pair-end 2 × 250 bp sequencing was performed using the Illumina NovaSeq platform (Illumina Inc., San Diego, CA, USA) at Panomic Biomedical Technology Co. (Suzhou, China).

### 2.10. Statistical Analysis

The experimental data were initially collated using Excel 2016 (version 2306, Microsoft, Redmond, WA, USA) software and subjected to one-way ANOVA using the general linear model (GLM) procedure with SAS 9.4 software (SAS Institute Inc., Cary, NC, USA). Multiple data comparisons were performed using the Tukey method, with differences considered significant when *p* ≤ 0.05 and highly significant when *p* ≤ 0.01; values of 0.05 < *p* < 0.1 were considered a significant trend.

## 3. Results

### 3.1. Performance

As shown in Table 2, the end weight, ADG and total DMI were higher in the group fed CAP than in the control group, though these values were not significantly different (*p* > 0.05). The addition of CAP had a tendency to increase starter DMI (*p* = 0.058), though there was no significant difference in feed efficiency between the groups (*p* = 0.246). There was no significant effect of CAP addition on the body height, body length, body slant length and chest circumference of calves throughout the trial period (Table 3, *p* > 0.05).

### 3.2. Fecal Score

No calves died during the entire trial period. We defined calf diarrhea as a fecal score of ≥2 that occurred for more than 2 consecutive days, and all calves had diarrhea during the trial period. As seen in Figure 1, the addition of high doses of CAP significantly reduced calf fecal scores.

### 3.3. Serum Biochemical Parameters

As shown in Table 4, the addition of CAP increased serum BUN concentrations (*p* = 0.030) and decreased CR concentrations (*p* = 0.002), though it had no significant effect on other serum biochemical parameters (*p* > 0.05).

### 3.4. Serum Antioxidant Parameters

As shown in Table 5, GSH-PX concentrations were significantly higher in calves fed high doses of CAP (*p* = 0.041) than in the control group. Both doses of CAP significantly increased SOD concentrations (*p* = 0.003). Low-dose CAP significantly reduced serum MDA levels (*p* = 0.029).

### 3.5. Serum Immunity Parameters

As shown in Table 6, high doses of CAP significantly increased serum IgA (*p* = 0.021), IgG (*p* = 0.030), IgM (*p* = 0.029) and IL-10 (*p* < 0.001) concentrations and decreased HP levels (*p* = 0.026). Compared to the control group, there was a trend toward lower IL-1β, IL-6 and SAA concentrations in the CAP-fed group (0.05 < *p* < 0.1).

### 3.6. Fecal Fermentation Parameters

As seen in Table 7, feeding CAP did not have a significant effect (*p* > 0.05) on total fecal SCFA, acetic acid, propionic acid, butyric acid, isobutyric acid, isovaleric acid and valeric acid in calves.

### 3.7. Fecal Microbial Composition

#### 3.7.1. Alpha-Diversity Analysis and Beta-Diversity Analysis

Alpha-diversity analysis showed that for Chao1, the observed species, the Shannon and Simpson indices were not significant among the three groups (*p* > 0.05; Figure 2A). Beta-diversity analysis based on weighted UniFrac principal coordinate analysis (PCoA) and Adonis difference analysis showed no significant separation of fecal microbial communities among the groups of calves (*p* > 0.05; Figure 2B).

#### 3.7.2. Relative Abundance and Structure of Gut Microbiota at Phylum and Genus Levels

To further determine the effect of CAP feeding on the gut microbial structure of calves, the compositions of the fecal bacterial community at the phylum and genus levels were compared among the groups. At the phylum level, the largest mean relative abundances in all groups were *Firmicutes*, *Actinobacteria* and *Bacteroidetes* (>96%; Figure 3A, Table 8), with no statistical differences reported between the mean relative abundances of the top ten groups in each group (*p* > 0.05). At the genus level, the top three abundant genera were *Collinsella*, *Bacteroides* and *Faecalibacterium*. There was no significant difference between the mean relative abundance of the top ten groups in each group (*p* > 0.05; Figure 3B, Table 9), though feeding CAP had a tendency to increase the relative abundance of *Collinsella* (*p* = 0.058), while the mean relative abundance of *Collinsella* was 76.51% and 51.15% higher in the L and H groups, respectively, than in the C group.

#### 3.7.3. LEfSe Analysis

Linear discriminant analysis of effect size (LDA Effect Size, Lefse) allows direct analysis of differences at all taxonomic levels simultaneously, and it has the ability to drill down into different subgroups to pick out marker microbial taxa that behave consistently in different subgroups. In this study, Lefse analysis screened 12 differential bacteria, including one phylum, one order, two families and six genera (Figure 4). In group C, *Cyanobacteria*, *Chloroplast*, *Stramenopiles*, *Actinomycetales*, *Corynebacteriaceae*, *Enterococcaceae*, *Corynebacterium*, *Eggerthella Streptococcus*, *Enterococcus* and *Megasphaera* were the dominant bacteria. In group H, *Coprococcus* was the dominant bacterium.

## 4. Discussion

The feed intake and weaning weight of calves affect their productive performance after growing into nursing cows, and the body size parameters of calves are related to their weight and development. Fecal SCFA content is related to nutrient digestion and utilization. Previous studies of CAP in dairy cows have focused on nursing cows, and their effects on nursing cow performance have been inconsistent. Foskolos et al. [21] fed 250 mg/d of CAP to nursing cows, which showed a trend toward increased DMI. Oh et al. [9] fed 1.2% CAP at 0.25, 0.5 or 1.0 g/d, which had no effect on DMI, nutrient digestibility or rumen fermentation parameters in cows. In another study, feeding 6% CAP at 2.0 g/d to nursing cows had no effect on DMI, though it significantly reduced Milk/DMI [22]. The reasons that CAP affects nutrient digestion in dairy cows may be related to alterations in the rumen microbial ecology, which affect pancreatic digestive enzyme secretion, promote gastric emptying and affect leptin levels [8]. In addition, studies on CAP in humans and mice have shown that CAP promotes weight loss [23,24]. The mechanisms are mainly as follows: the promotion of fat oxidation, the inhibition of transcription of genes responsible for proteins that stimulate lipogenesis and accumulation and the stimulation of adrenaline and noradrenaline secretion by the adrenal glands to enhance thermogenesis [11]. The different findings may be related to the different species, the different ages of the test animals and the form and dose of CAP addition. In the present study, feeding CAP did not negatively affect all growth performance of nursing calves or fecal fermentation parameters, and there was a tendency to increase the intake of calves on starters. Serum BUN concentration reflects the level of protein anabolism in animals, and Blome et al. [25] reported that BUN concentration increased linearly with an increase in the crude protein content in the diet. CR is the end product of creatine and phosphocreatine metabolism in muscle tissue, and serum CR levels decreased linearly with increasing dietary energy levels [26]. In the present study, feeding CAP increased BUN and decreased CR levels, which corresponds to the trend of increasing DMI in starter diets.

The nursing calves have immature immune systems and poor autoimmunity, making them susceptible to stress and disease early in life. In this study, the experiment was conducted at 0–1 months of age, when the incidence of calf diarrhea is highest, to observe the effect of CAP addition on the incidence of calf diarrhea and serum parameters. During the trial, all calves experienced diarrhea. However, calves fed high doses of CAP had lower fecal scores and daily incidences of diarrhea. Antioxidant function is an important indicator of the health status of animals. The antioxidant defense system of the body includes enzymatic and non-enzymatic systems. The enzymatic antioxidant system is mainly composed of GSH-Px and SOD [27]. MDA is one of the most common indicators of lipid peroxidation. When stress occurs in the body, a large number of free radicals are generated, which induce the peroxidation of polyunsaturated fatty acids in the biofilm to produce MDA [28]. In this study, feeding CAP increased serum GSH-Px and SOD levels and decreased MDA levels, and the effect was greater at higher doses. CAP was found to inhibit lipid peroxidation in human erythrocyte membranes and liver peroxidation in rats [13,29]. In addition, CAP reduced oxidative stress markers, such as thiobarbituric acid-reactive substances and MDA, in rat serum and liver, lung, kidney and muscle [30,31]. CAP was also found to scavenge 1,1-diphenyl-2-trinitrophenylhydrazine (DPPH) radicals [32]. Cunha et al. showed that the addition of 0.5% capsaicin to the diet of nursing sheep increased SOD levels [33]. The results of the above studies were similar to those of the present study. Serum concentrations of IgA, IgG and IgM are important indicators of the body’s immune function. Among these factors, serum IgA is the main antibody of the exocrine fluid, which prevents pathogens from adhering to the mucosa to defend against mucosal infections and acts as a suppressor of the inflammatory response. Its concentration decreases during the onset of diarrhea [34]. In the present study, the addition of CAP to MR in calves significantly increased serum IgA, IgG and IgM concentrations and showed a dose dependence. These findings explain the lower fecal scores in the treated group. The pro-inflammatory cytokines IL-1β, IL-6 and TNF-α can enhance systemic or local inflammatory responses, aggravate histopathology and disrupt intestinal barrier function, leading to the development of enterocolitis, while the anti-inflammatory factor IL-10 has a negative feedback regulatory effect and can reduce the production and release of pro-inflammatory cytokines [35]. In this study, feeding CAP significantly increased IL-10 concentrations and had a tendency to decrease IL-1β and IL-6. SAA and HP are two major bovine acute phase proteins (APP), which are a group of stress proteins that regulate the inflammatory response, in vivo immune function and recovery mechanisms; their concentrations increase when animals are exposed to stress, such as inflammation and infectious diseases. Therefore, APP can be used as a biomarker to assess the degree of inflammatory response in animals subjected to stress [36]. Studies have shown that serum biomarker concentrations of pro-inflammatory cytokines, amyloid and binding bead protein were significantly higher in diseased calves than in healthy calves [37]. In this study, calves supplemented with high doses of CAP contained significantly lower levels of HP and a trend toward lower levels of IL-1β, IL-6 and SAA, thus explaining the lower fecal scores in the treated group of calves. Oh et al. [9] found that CAP increased neutrophil activity and immune cells associated with the acute-phase immune response in cows, which is similar to the results of the present study. In summary, feeding 0.3 mL/day of CAP improves the antioxidant and immune capacity of calves to resist disease and reduces the inflammatory response required to maintain immune system homeostasis, thereby reducing the incidence of calf diarrhea.

Animal gut microbiota are closely related to the physical health condition of animals [38]. Alterations in gut microbiota can cause a decrease in host immunity and increased susceptibility to disease [39]. To study the effect of CAP on the fecal gut microbiota of calves, fecal samples were collected on the last day of the experiment to enable sequencing analysis. The results showed that there was a tendency for CAP to increase *Collinsella* at the genus level. In a comparative study of fecal micro-organisms in healthy and diarrheic calves, *Collinsella* abundance was found to be significantly higher in healthy calves than in diarrheic calves, while *Collinsella* was found to be significantly associated with rsodeoxycholic acid (UDCA) production via combined metabolomic and microbiomic analysis [40]. UDCA is a typical “therapeutic” bile acid that has been shown, in previous studies, to alleviate dextran sulfate sodium-induced colitis and *C. difficile* infection, as well as to have an excellent effect on protecting colonic epithelial cells from oxidative damage and apoptosis [41]. Thus, higher *Collinsella* abundance in calves in the treated group was beneficial in preventing and relieving diarrhea. Related studies have reported that CAP has an antibacterial effect, inhibits Escherichia coli and Staphylococcus aureus [42], and has the ability to reduce the virulence of Staphylococcus aureus by inhibiting its efflux pump and reducing its invasive capacity [43]. In addition, CAP reduces the number of H. pylori and reduces H. pylori pathogenicity in a dose-dependent manner [44]. In this study, Lefse analysis revealed that CAP reduced the abundance of five genera—*Corynebacterium*, *Eggerthella*, *Streptococcus*, *Enterococcus* and *Megasphaera*—as well as *Enterobacteriaceae*. *Corynebacterium* is a secondary pathogen of mastitis in dairy cows. Correlation analysis showed that *Corynebacterium* was positively correlated with the expression of TNF-α [45]. *Eggerthella* is a Gram-positive genus associated with severe gastrointestinal pathology [46]. Xin et al. [47] found that the relative abundance of *Eggerthella* was significantly higher in diarrheic calves than in healthy calves. The shift in the GI microbiota from specific to parthenogenic anaerobic bacteria is considered to be a marker of dysbiosis in the GI microbiota, and the proliferation of parthenogenic anaerobic bacteria is closely associated with pro-inflammatory mucosal immune responses and plays a pathogenic role in GI inflammation [48]. Gomez et al. [49] found that *Streptococcus*, *Enterococcus* and other pathogenic parthenogenic anaerobic bacteria were enriched in diarrheic calves. The above-mentioned diarrhea-causing harmful bacteria were also enriched in calves in group C that were not fed CAP, explaining the higher incidence of abnormal fecal scores in calves in group C. Studies have shown that CAP inhibits bacteria in the animal intestine and rumen of ruminants [50]. The antimicrobial effect of CAP is highly correlated with the concentration used, and it inhibits the expression of genes related to bacterial cell growth [51]. In addition, the phenolic group of CAP is hydrophobic and can inhibit bacterial growth by reducing membrane stability [52]. Orndorff et al. [53] showed that the addition of CAP to the diet reduced the incidence of Salmonella infection in broilers. In the present study, feeding CAP to calves significantly reduced the enrichment of harmful bacteria in diarrhea and benefited the gastrointestinal health of calves. *Megasphaera* is an anaerobic bacterium that can use lactic acid as a growth substrate. It is usually found in the rumen, and the genus has a very low abundance in the rumen of calves [54]. No reports have been seen regarding the presence of *Megasphaera* in the rectal fecal microbiota of dairy cows. *Coprococcus* is involved in the carbohydrate catabolic utilization process, and its abundance is positively correlated with carbohydrate supplementation [55]. Its enrichment in calves in group H corresponded to the higher total DMI recorded in group H. *Enterobacteriaceae* is a family of Gram-negative bacilli that includes many pathogenic bacterial genera, of which *Escherichia coli* and *Salmonella spp*. are among the most important bacterial causes of diarrhea-related death in calves [56]. In conclusion, the addition of CAP can inhibit the proliferation of the harmful bacteria *Corynebacterium*, *Eggerthella*, *Streptococcus*, *Enterococcus* and *Enterobacteriaceae* in the gut and maintain microecological balance in the gut.

## 5. Conclusions

In conclusion, the addition of high doses of CAP had no negative effect on calf growth performance; increased serum BUN, GSH-PX, SOD, IgA, IgG, IgM levels and IL-10; and decreased CR, MDA and HP levels. Therefore, the results of this study suggest that 0.3 mL/day CAP could improve the health of calves by increasing antioxidant and immune capacity and reducing the abundance of harmful gut bacteria.

## Figures and Tables

**Figure 1 animals-13-02309-f001:**
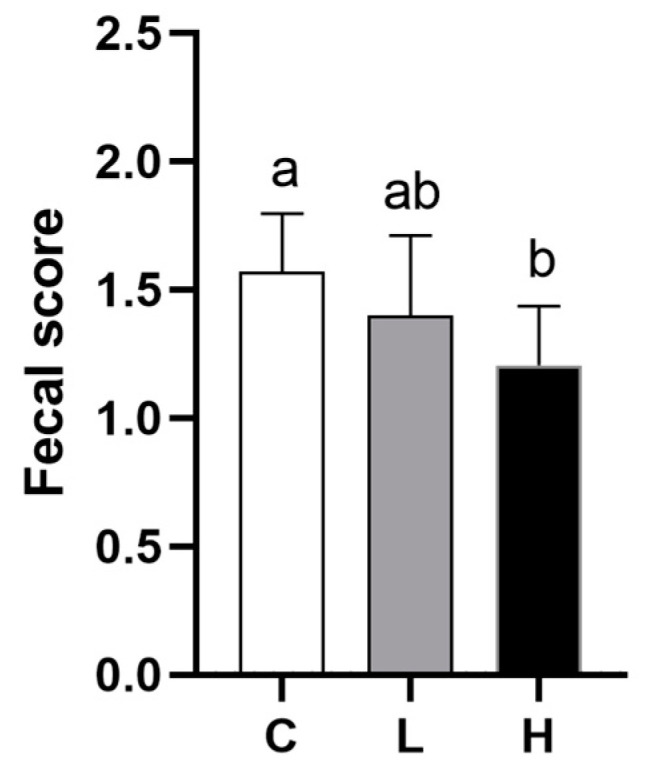
Fecal scores of Holstein calves fed either the control (C) or supplemented diet with low-level capsaicin (L) or the supplemented diet with high-level capsaicin (H). Values with unlike lowercase letters differed significantly (*p* < 0.05).

**Figure 2 animals-13-02309-f002:**
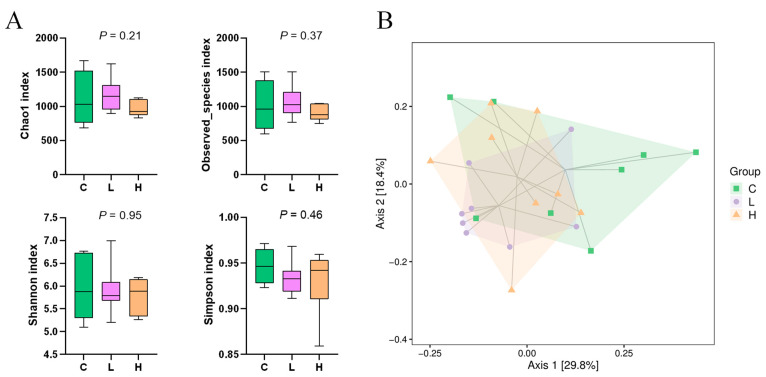
Alpha-diversity indices (**A**) and weighted UniFrac-based PCoA plot (**B**) of fecal samples of Holstein calves fed either the control (C) or supplemented diet with low-level capsaicin (L) or the supplemented diet with high-level capsaicin (H).

**Figure 3 animals-13-02309-f003:**
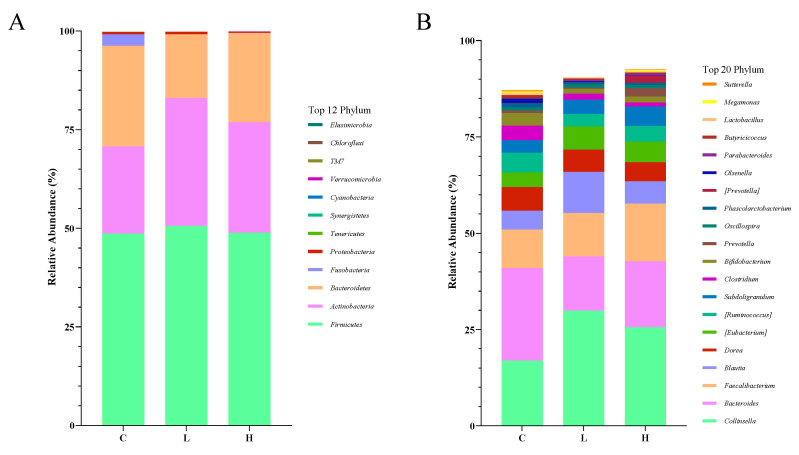
The composition of gut microbiota on phylum (**A**) and genus (**B**) level of fecal samples of Holstein calves fed either the control (C) or supplemented diet with low-level capsaicin (L) or the supplemented diet with high-level capsaicin (H).

**Figure 4 animals-13-02309-f004:**
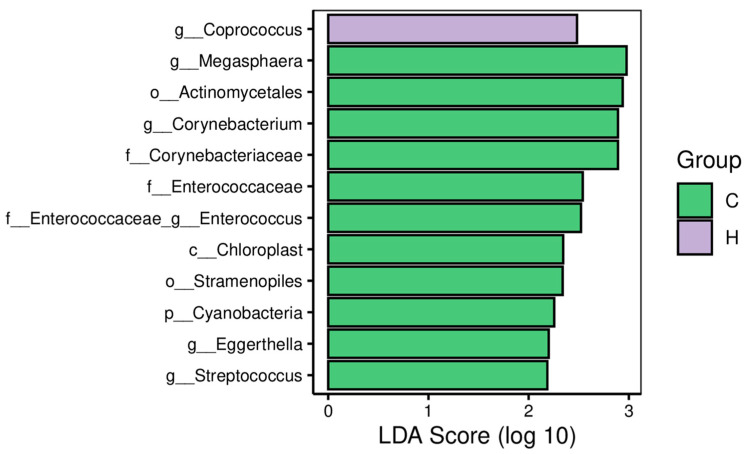
Differential bacteria obtained via Lefse analysis of Holstein calves fed either the control (C) or supplemented diet with low-level capsaicin (L) or the supplemented diet with high-level capsaicin (H). The thresholds of LDA are > 2 and *p* < 0.05.

**Table 1 animals-13-02309-t001:** Chemical composition (% of DM, unless otherwise noted) of calf starter and milk replacer.

Item	Calf Starter ^1^	Milk Replacer ^2^
DM (%)	88.5	94.8
Ash	9.6	9.1
CP	20.9	20.2
EE	5.3	16.2
NDF	21.9	-
ADF	11.0	-
NFC	42.3	54.5

^1^ Commercial calf starter (Guangdong Wen’s Dairy Industry Co., Zhaoqing, China). ^2^ Milk replacer (Nutrifeed, Veghel, The Netherlands).

**Table 2 animals-13-02309-t002:** Body weight, ADG and feeding data of Holstein calves fed either the control (C) or supplemented diet with low-level capsaicin (L) or the supplemented diet with high-level capsaicin (H) ^1^.

Item	Treatment			SEM	*p*-Value ^2^
	C	L	H		
BW, kg					
S0	38.36	39.44	38.51	1.327	0.826
S1	64.31	66.19	66.40	1.910	0.700
ADG, kg/d	0.92	0.89	0.93	0.027	0.562
Total DMI ^3^, g/d	1378.58	1445.33	1452.80	26.539	0.118
Starter DMI, g/d	83.85	102.08	94.67	5.061	0.058 ^#^
Feed efficiency ^4^	0.65	0.62	0.64	0.015	0.246

^1^ S0 = first day of the trial (5 d), S1 = end of stage 1 (35 ± 2.04 d). ^2^ The # sign represents the trend of 0.05 *< p <* 0.1. ^3^ Starter and milk replacer DMI. ^4^ Feed efficiency = ADG (g/d)/total intake (g/d).

**Table 3 animals-13-02309-t003:** Body size parameters of Holstein calves fed either the control (C) or supplemented diet with low-level capsaicin (L) or the supplemented diet with high-level capsaicin (H) ^1^.

Item	Treatment			SEM	*p*-Value
	C	L	H		
Body height					
S0, cm	77.96	77.91	78.45	0.607	0.788
S1, cm	87.43	86.73	87.56	0.686	0.660
Change, cm/d	0.30	0.33	0.31	0.055	0.504
Body length					
S0, cm	59.50	59.55	60.45	1.223	0.828
S1, cm	76.76	77.41	78.13	0.794	0.491
Change, cm/d	0.58	0.60	0.59	0.126	0.940
Body slant length					
S0, cm	61.58	60.94	62.36	1.303	0.748
S1, cm	78.30	79.04	80.50	0.796	0.163
Change, cm/d	0.56	0.61	0.61	0.135	0.716
Chest					
S0, cm	78.38	77.51	78.53	1.065	0.771
S1, cm	89.91	90.63	90.35	1.047	0.890
Change, cm/d	0.39	0.44	0.40	0.062	0.250

^1^ S0 = first day of the trial (5 d), S1 = end of stage 1 (35 ± 2.04 d).

**Table 4 animals-13-02309-t004:** Blood biochemical parameters of Holstein calves fed either the control (C) or supplemented diet with low-level capsaicin (L) or the supplemented diet with high-level capsaicin (H) ^1^.

Item	Treatment			SEM	*p*-Value
	C	L	H		
GLU, mmol/L	8.55	7.72	8.53	0.333	0.522
ALT, U/L	14.41	13.55	13.91	0.950	0.934
AST, U/L	39.36	45.86	47.83	1.674	0.121
TP, g/L	57.24	53.81	54.03	0.803	0.173
ALB, g/L	27.37	27.93	26.57	0.264	0.132
GLOB, g/L	29.87	25.88	27.46	0.788	0.139
ALP, U/L	336.53	351.60	368.82	22.07	0.838
BUN, mmol/L	1.43 ^b^	1.77 ^a^	1.73 ^ab^	0.054	0.030
CR, μmol/L	86.87 ^a^	73.90 ^b^	78.35 ^b^	1.332	0.002
BHBA, mmol/L	0.05	0.05	0.05	0.003	0.828

^1^ The values (row) with unlike lowercase letters differed significantly (*p* < 0.05).

**Table 5 animals-13-02309-t005:** Blood antioxidant parameters of Holstein calves fed either the control (C) or supplemented diet with low-level capsaicin (L) or the supplemented diet with high-level capsaicin (H) ^1^.

Item	Treatment			SEM	*p*-Value
	C	L	H		
T-AOC, mmol/L	0.68	0.70	0.68	0.008	0.684
GSH-PX, U/ml	75.41 ^b^	83.54 ^ab^	89.33 ^a^	2.091	0.041
SOD, U/ml	140.55 ^b^	150.54 ^a^	149.07 ^a^	1.107	0.003
MDA-nmol/ml	3.53 ^a^	2.49 ^b^	3.00 ^ab^	0.146	0.029

^1^ The values (row) with unlike lowercase letters differed significantly (*p* < 0.05).

**Table 6 animals-13-02309-t006:** Blood immunity parameters of Holstein calves fed either the control or supplemented diet with low-level capsaicin (L) or the supplemented diet with high-level capsaicin (H) ^1^.^.^

Item	Treatment			SEM	*p*-Value
	C	L	H		
IgA, μg/mL	974.58 ^b^	1343.93 ^ab^	1402.43 ^a^	62.138	0.021
IgG, mg/mL	4.70 ^b^	5.32 ^ab^	5.70 ^a^	0.143	0.030
IgM, μg/mL	786.32 ^b^	964.39 ^ab^	1041.02 ^a^	36.699	0.029
IL-1β, pg/mL	444.83	357.99	386.82	14.469	0.072
IL-6, pg/mL	108.07	102.23	97.31	1.749	0.063
IL-10, pg/mL	49.14 ^b^	48.30 ^b^	63.40 ^a^	1.545	<0.001
TNF-α, pg/mL	89.52	88.85	87.34	3.321	0.963
SAA, μg/mL	7.86	6.99	6.74	0.210	0.096
HP, μg/mL	24.61 ^a^	24.02 ^a^	20.40 ^b^	0.631	0.026

^1^ The values (row) with unlike lowercase letters differed significantly (*p* < 0.05).

**Table 7 animals-13-02309-t007:** Fecal SCFA content of Holstein calves fed either the control (C) or supplemented diet with low-level capsaicin (L) or the supplemented diet with high-level capsaicin (H).

Item	Treatment			SEM	*p*-Value
	C	L	H		
Total SCFA, mmoL/L	50.20	65.06	48.80	3.941	0.184
Acetic acid, %	61.90	62.24	61.28	0.014	0.958
Propionic acid, %	20.55	20.90	21.51	0.013	0.957
Butyric acid, %	13.03	12.98	11.41	0.005	0.371
Isobutyric acid, %	1.42	1.09	1.14	0.002	0.816
Isovaleric acid, %	2.02	1.82	1.67	0.003	0.894
Valeric acid, %	1.77	1.42	1.81	0.027	0.953

**Table 8 animals-13-02309-t008:** Top five relative abundances of gut microbiota at the phylum level of Holstein calves fed either the control diet (C) or the supplemented diet with low-level capsaicin (L) or the supplemented diet with high-level capsaicin (H).

Item	Treatment			SEM	*p*-Value
	C	L	H		
*Firmicutes*	48.61	50.60	48.82	0.031	0.959
*Actinobacteria*	22.12	32.43	28.07	0.021	0.158
*Bacteroidetes*	25.52	16.08	22.57	0.025	0.297
*Fusobacteria*	2.88	0.00	0.01	0.007	0.172
*Proteobacteria*	0.58	0.67	0.29	0.001	0.282

**Table 9 animals-13-02309-t009:** Top ten relative abundances of gut microbiota at the genus level of Holstein calves fed either the control (C) or supplemented diet with low-level capsaicin (L) or the supplemented diet with high-level capsaicin (H).

Item	Treatment			SEM	*p*-Value ^1^
	C	L	H		
*Collinsella*	16.95	29.91	25.62	0.021	0.058 ^#^
*Bacteroides*	23.92	14.08	17.11	0.023	0.235
*Faecalibacterium*	10.08	11.20	14.99	0.017	0.465
*Blautia*	4.84	10.72	5.78	0.013	0.161
*Dorea*	6.19	5.79	4.95	0.008	0.814
*[Eubacterium]*	3.76	6.01	5.31	0.013	0.764
*[Ruminococcus]*	5.12	3.26	4.10	0.009	0.709
*Subdoligranulum*	3.29	3.70	5.01	0.012	0.839
*Clostridium*	3.76	1.51	1.02	0.009	0.394
*Bifidobacterium*	3.27	1.30	1.60	0.004	0.129

^1^ The # sign represents the trend of 0.05 *< p <* 0.1.

## Data Availability

The raw sequence data reported in this paper have been deposited in the Genome Sequence Archive (Genomics, Proteomics and Bioinformatics 2021) at the National Genomics Data Center (Nucleic Acids Res 2022), China National Center for Bioinformation/Beijing Institute of Genomics, Chinese Academy of Sciences (GSA: CRA011263), and they are publicly accessible at https://ngdc.cncb.ac.cn/gsa/ (accessed on 2 June 2023).

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
