# Peer review of "The Effect of Capsaicin on Growth Performance, Antioxidant Capacity, Immunity and Gut Micro-Organisms of Calves"

_animals, 2023, doi:10.3390/ani13142309_

Round 1

Reviewer 1 Report

This manuscript mainly considers the effect of capsaicin on growth performance, antioxidant capacity, immunity, and gut microorganisms of calves. This is a well-written paper containing interesting results. For the benefit of the reader, however, a number of points need clarifying and certain statements require further justification.
1, What is the definition of high dose and low dose capsaicin? Whether there is literature support?

2, What is the CV value measured by Elisa?
3, The supplemental amount of capsaicin was only 0.15 and 0.3 ml per day, and the blood physiology and gastrointestinal flora of calves changed greatly during growth and development. Compared with 10 L/d of milk replacer, whether the supplemental amount was insufficient to show that the changes in parameters in this study was caused by the addition of capsaicin.
4, The feeding time of 5 days of age is relatively early, and whether the calves have anorexia or resistance to the milk replacer with capsaicin?
5, The whole experiment period was only one month. Did you consider tracking the fecal and rumen fermentation levels of calves during 2-6 months of age?

6, Whether the dairy cows were changed throughout the experiment. Whether abnormal circumstance occur during the feeding process.

7, Try to set the problem discussed in this manuscript in more clear.

8. Immunoglobulin A (IgA), Immunoglobulin G (IgG) and Immunoglobulin M 130 (IgM) were determined by ELISA according to the kit instructions (Beijing Kyushu  Taikang Biotechnology Co., Ltd.). Interleukin-1β (IL-1β), interleukin-6 (IL-6), interleukin-132 10 (IL-10), tumor necrosis factor-α (TNF-α), amyloid A (SAA) and binding bead protein (HP) were determined by ELISA according to the kit instructions (Shanghai Zhangshi Biotechnology Co., Ltd.). What is the website for the Beijing Kyushu Taikang Biotechnology Co., Ltd. and Shanghai Zhangshi Biotechnology Co., Ltd.? Because I do not find these website.

Author Response

The content of the attachment:

Replies to the editor’s and reviewer’s comments

animals-2472245  

“The Effect of Capsaicin on Growth Performance, Antioxidant Capacity, Immunity and Gut Microorganisms of Calves”

Thank you very much for your letter dated, and the reviewers’ good suggestions. Based on your comment and request, we have modified the original manuscript. Here, we attached the revised manuscript in the formats of MS word, for your approval. A document answering every question from the reviews was also summarized and enclosed.

A revised manuscript with the correction sections yellow marked in the manuscript for easy check purposes.

Should you have any questions, please contact us without any hesitation.

Reviewer 1’s Comments:

Comment 1: What is the definition of high dose and low dose capsaicin? Whether there is literature support?

Response: Thanks very much. As described previously (Vittorazzi et al. 2022), supplementation of Holstein cows (660 ± 85.9 kg BW) with 0.75 g/d or 1.5 g/d of capsaicin [contains 10 g/kg (minimum) of encapsulated pepper] resulted in an increase in dry matter intake in the cows. Cows fed 1.5 g/d of capsaicin tended to have greater dry matter intake than cows fed 0.75 g/d of capsaicin. And a tendency toward greater milk protein content was observed for cows fed 1.5 g/d of capsaicin than 0.75 g/d of capsaicin. Also, note that capsaicin doses greater than 1.5 g/d did not show positive effects and even reduced cow performance (Oh et al., 2013). Therefore, converting to calves, our study defined a capsaicin additive amount of 0.15 ml/d as a low dose and positioned a capsaicin additive amount of 0.3 ml/d as a high dose.

  1. Vittorazzi, P.C.; Takiya, C.S.; Nunes, A.T.; Chesini, R.G.; Bugoni, M.; Silva, G.G.; Silva, T.B.P.; Dias, M.S.S.; Grigoletto, N.T.S.; Rennó, F.P. Feeding encapsulated pepper to dairy cows during the hot season improves performance without affecting core and skin temperature. J. Dairy Sci. 2022, 105, 9542-9551, doi:https://doi.org/10.3168/jds.2022-22078.
  2. Oh, J.; Hristov, A.N.; Lee, C.; Cassidy, T.; Heyler, K.; Varga, G.A.; Pate, J.; Walusimbi, S.; Brzezicka, E.; Toyokawa, K.; et al. Immune and production responses of dairy cows to postruminal supplementation with phytonutrients. J. Dairy Sci. 2013, 96, 7830-7843, doi:https://doi.org/10.3168/jds.2013-7089.

Comment 2: What is the CV value measured by Elisa?

Response: Thanks very much. The CV value for the measured parameters by ELISA assay in the C, L, and H groups were as follows: 18%, 21%, and 11% for IL-1β; 2%, 13%, and 4% for IL-6; 13%, 21%, and 5% for IL-10; 19%, 10%, and 21% for TNF-α; 13%, 12%, and 15% for SAA; 13%, 15%, and 6% for HP; 21%, 23%, and 23% for IgA; 12%, 16%, and 10% for IgG; and 9%, 21%, and 19% for IgM. Although some of the CV values were relatively high, the SAS procedure was used to plot box plots for all data in this study to determine if there were outliers and none of the above indicators were determined to have outliers beyond the upper and lower boundaries of the box plots.

Comment 3: The supplemental amount of capsaicin was only 0.15 and 0.3 ml per day, and the blood physiology and gastrointestinal flora of calves changed greatly during growth and development. Compared with 10 L/d of milk replacer, whether the supplemental amount was insufficient to show that the changes in parameters in this study was caused by the addition of capsaicin.

Response: Thanks very much. As described previously (Vittorazzi et al. 2022), supplementation of Holstein cows (660 ± 85.9 kg BW) with 0.75 g/d or 1.5 g/d of capsaicin [contains 10 g/kg (minimum) of encapsulated pepper] can improve yield of fat-corrected milk and milk solids by increasing feed intake without affecting nutrient digestibility and body temperature of lactating cows during the hot season. Other studies (Oh et al. 2015) found 250, 500, or 1,000 mg of capsaicin/cow increased neutrophil activity and immune cells associated with the acute phase immune response in cows. The first of the above studies added capsaicin to the concentrate and the second added to the TMR. Therefore it is also reasonable to add 0.15 ml or 0.3 ml of capsaicin to 5 L /meal of milk replacer in this study.

  1. Vittorazzi, P.C.; Takiya, C.S.; Nunes, A.T.; Chesini, R.G.; Bugoni, M.; Silva, G.G.; Silva, T.B.P.; Dias, M.S.S.; Grigoletto, N.T.S.; Rennó, F.P. Feeding encapsulated pepper to dairy cows during the hot season improves performance without affecting core and skin temperature. J. Dairy Sci. 2022, 105, 9542-9551, doi:https://doi.org/10.3168/jds.2022-22078.
  2. Oh, J.; Giallongo, F.; Frederick, T.; Pate, J.; Walusimbi, S.; Elias, R.J.; Wall, E.H.; Bravo, D.; Hristov, A.N. Effects of dietary Capsicum oleoresin on productivity and immune responses in lactating dairy cows. J Dairy Sci 2015, 98, 6327-6339, doi:10.3168/jds.2014-9294.

Comment 4: The feeding time of 5 days of age is relatively early, and whether the calves have anorexia or resistance to the milk replacer with capsaicin?

Response: Thanks very much. Since the capsaicin used in this study was unencapsulated liquid capsaicin, this study had been pre-tested before the formal trial and finally concluded that none of the calves were anorexic or resistant when 0.3 ml of capsaicin was added to 1.5 L of milk replacer.

Comment 5: The whole experiment period was only one month. Did you consider tracking the fecal and rumen fermentation levels of calves during 2-6 months of age?

Response: Thanks very much. In this particular study, our focus was on investigating the immediate effects of capsaicin supplementation on the growth and health of calves within a one-month period. However, we acknowledge the importance of examining the long-term impacts of capsaicin on calves at different stages of growth. Our future research plans include investigating the influence of capsaicin on the growth and health of calves at older ages. We recognize that this is an essential aspect that needs to be addressed, and it will be a part of our subsequent studies.

Comment 6: Whether the dairy cows were changed throughout the experiment. Whether abnormal circumstance occur during the feeding process.

Response: Thanks very much. In this study, we ensured that the same group of dairy calves was used throughout the entire experimental period. There were no changes in the composition of the calves group, which allowed us to maintain consistency in our observations and data collection. Furthermore, we are pleased to report that no abnormal circumstances, such as aversion or resistance to feeding, were observed during the feeding process. On the contrary, our results indicated a trend of increased dry matter intake with the addition of capsaicin, suggesting a positive response to the supplementation.

Comment 7: Try to set the problem discussed in this manuscript in more clear.

Response: Thanks very much. In response to your comments, I have made appropriate minor changes to the discussion section to make my presentation clearer, more concise, and more explicit. (lines 311-433).

Comment 8: Immunoglobulin A (IgA), Immunoglobulin G (IgG) and Immunoglobulin M 130 (IgM) were determined by ELISA according to the kit instructions (Beijing Kyushu  Taikang Biotechnology Co., Ltd.). Interleukin-1β (IL-1β), interleukin-6 (IL-6), interleukin-132 10 (IL-10), tumor necrosis factor-α (TNF-α), amyloid A (SAA) and binding bead protein (HP) were determined by ELISA according to the kit instructions (Shanghai Zhangshi Biotechnology Co., Ltd.). What is the website for the Beijing Kyushu Taikang Biotechnology Co., Ltd. and Shanghai Zhangshi Biotechnology Co., Ltd.? Because I do not find these website.

Response: Thanks very much. First of all, about Beijing Kyushu Taikang Biotechnology Co., Ltd., you may not find its website because my translation is different from the English name of the company. The correct name should be BGI Jiuzhou Taikang Biotechnology Co., Ltd, and its website is https://bgi-jztk.com/sy. I have corrected the correct company name in the revised version (line 130). Also for Shanghai Zhangshi Biotechnology Co., Ltd., its website is http://www.shzhshbio.com/.

Thank you and all the reviewers for the kind comments.

Sincerely yours,

Minqiang Su

Reviewer 2 Report

Abstract

1.the genus Capsicum.. In italics  :  genus Capsicum

3. Immunity: change to “immunity”

4. Include summary of material and methods

5. higher dose of capsaicin , please write what these doses

Introduction

1 Change Antibiotics to “antimicrobial agents” or antibacterial

2. Capsicum annuum is an herb of the genus Capsicum and all names of Bacteria

Please write in italic font Capsicum annuum is an herb of the genus Capsicum

3. line 53-- Therefore, the aim of this study was 52 to evaluate the effects of adding different doses of CAP on growth- change to ‘Therefore, tha aim in this study”

Material and Methods

1Line 71 “ 8 groups of 3 calves change to Eight groups of three calves.¨

Results

Figure 3. “gut microflora” or gut flora the term microflora or flora are not correct

Change In all text to “ gut microbiota “ -

  Please increase font size for phyla and genera names

Results and Discussion

Please include comments about the Enterobacteriaceae - with special emphasis on the Escherichia coli and Salmonella spp.

References about the antimicrobial effects of  Capsaicin can be included in the discussion

the data were very well felt and organized. The results are important to find alternatives to improve the health of young calves and contribute to reducing the use of antimicrobials.

Abstract

1.the genus Capsicum.. In italics  :  genus Capsicum

3. Immunity: change to “immunity”

4. Include summary of material and methods

5. higher dose of capsaicin , please write what these doses

Introduction

1 Change Antibiotics to “antimicrobial agents” or antibacterial

2. Capsicum annuum is an herb of the genus Capsicum and all names of Bacteria

Please write in italic font Capsicum annuum is an herb of the genus Capsicum

3. line 53-- Therefore, the aim of this study was 52 to evaluate the effects of adding different doses of CAP on growth- change to ‘Therefore, tha aim in this study”

Material and Methods

1Line 71 “ 8 groups of 3 calves change to Eight groups of three calves.¨

Results

Figure 3. “gut microflora” or gut flora the term microflora or flora are not correct

Change In all text to “ gut microbiota “ -

  Please increase font size for phyla and genera names

Results and Discussion

Please include comments about the Enterobacteriaceae - with special emphasis on the Escherichia coli and Salmonella spp.

References about the antimicrobial effects of  Capsaicin can be included in the discussion

Author Response

The content of the attachment.

Reviewer 2’s Comments:

Abstract

Comment 1: the genus Capsicum. In italics : genus Capsicum

Response: Thanks very much. Revised as you suggested. (line 12)

Comment 2: Immunity: change to “immunity”

Response: Thanks very much. Revised as you suggested. (line 14)

Comment 3: Include summary of material and methods

Response: Thanks very much. A summary of the materials and methods has been included in the revised manuscript. (lines 15-20)

Comment 4: higher dose of capsaicin , please write what these doses

Response: Thanks very much. A specific dose (0.3 ml/d) has been added to the revised manuscript. (line 22)

Introduction

Comment 1: Change Antibiotics to “antimicrobial agents” or antibacterial

Response: Thanks very much. The word "Antibiotics" has been changed to "antibacterial" in the revised manuscript. (line 42, 44, 45, 46)

Comment 2:  Capsicum annuum is an herb of the genus Capsicum and all names of Bacteria

Please write in italic font Capsicum annuum is an herb of the genus Capsicum

Response: Thanks very much. Revised as you suggested. (line 48)

Comment 3: line 53-- Therefore, the aim of this study was 52 to evaluate the effects of adding different doses of CAP on growth- change to ‘Therefore, tha aim in this study”

Response: Thanks very much. Revised as you suggested. (line 60)

Material and Methods

Comment 1: 1Line 71 “ 8 groups of 3 calves change to Eight groups of three calves.¨

Response: Thanks very much. Revised as you suggested. (line 79)

Results

Comment 1: Figure 3. “gut microflora” or gut flora the term microflora or flora are not correct

Response: Thanks very much. All errors have been corrected to "gut microbiota" in the full text of the revised manuscript. (line 287)

Comment 2: Change In all text to “ gut microbiota “ -

Response: Thanks very much. Revised as you suggested. (lines 287, 383, 384, 385, 407, 425)

Comment 3: Please increase font size for phyla and genera names

Response: Thanks very much. The drawing has been redrawn according to your suggestion. (line 286)

Results and Discussion

Comment 1: Please include comments about the Enterobacteriaceae - with special emphasis on the Escherichia coli and Salmonella spp.

Response: Thanks very much. The corresponding content has been added to the revised manuscript. (lines 402, 428-430)

Comment 2: References about the antimicrobial effects of  Capsaicin can be included in the discussion

Response: Thanks very much. The corresponding content has been added to the revised manuscript. (lines 396-400)

Thank you and all the reviewers for the kind comments.

Sincerely yours,

Minqiang Su

Reviewer 3 Report

This is an intervention study assessing the impact of different capsaicin levels on various calf parameters. It is a very thorough study utilising a wide range of diagnostic and lab techniques.

Line 14 - immunity doesn't need a capital letter.

Line 29 - i am not sure the term 'poor autoimmunity' is biologically correct. This is suggestive of an immune disfunction, which is not the case with neonatal calf immunity. Please change this phrase.

Line 33 - lactating calves implies the calves are producing milk through the acct of lactating, rather than consuming milk. Please re-phrase this.

Line 46 - do these bacteria names need to be in Italics?

Line 58 - how big was the farm, how many animals did it have?

Line 67 - what is a colostrum baster?

Line 71 - do not start a sentence with a number. Eight rather than 8.

Line 78 - experiment rather than experimental

Line 84 - exactly how much milk replacer powder was fed in g? It is unusual to give a ration of 1:6 for milk replacer.

Line 87 - the grammar of this sentence needs adjusting.

Line 108 - how were calves weighed?

Line 118 - grammar of this sentence needs improving. Was this a commercial preparation of oral fluids? How were the fluids offered?

Line 157 - grammar of this sentence needs improving.

For section 2.8 and 2.9, much of the methodology in this section could be presented either in a flow chart of table to make it easier to follow.

Line 193 - does not need a hyphen between soft-ware

Table 2 - highlight the p-value for starter DMI as being P<0.1 for a trend. This is usually done with a # or similar mark.

Line 217 - calf not calve

Figure 1 - the use of the lower case letter is confusing. Please change this to some other method of identifying differences. This also applies to the tables. This needs changing - what is the difference between a and ab and b?

Line 233 - does medium dose refer to the low dose of capsaicin?

Line 255 - does this mean there is no significant difference between the groups?

Line 299-314 - some of this information should be moved up to the introduction section of the article as it is background to how Capsaicin has been used before.

Line 321 - space missing  between diet. CR

Line 326 - change the term lactating calves as mentioned above

Line 390 - Does xin need a capital letter, and why is the reference repeated at the end of the sentence?

Line 400 - Capital letter for The

Line 412 -  re-phrase the 'in conclusion' here as this sentence is followed by the papers conclusion.

Minor edits needed to improve grammer.

Author Response

The content of the attachment.

Reviewer 1’s Comments:

Comment 1: Line 14 - immunity doesn't need a capital letter.

Response: Thanks very much. Revised as you suggested. (line 14)

Comment 2: Line 29 - i am not sure the term 'poor autoimmunity' is biologically correct. This is suggestive of an immune disfunction, which is not the case with neonatal calf immunity. Please change this phrase.

Response: Thanks very much. It has been revised to "immature immune response" in the revised manuscript. (line 35)

Comment 3: Line 33 - lactating calves implies the calves are producing milk through the acct of lactating, rather than consuming milk. Please re-phrase this.

Response: Thanks very much. It has been changed to "nursing calves" in the revised manuscript. (line 39)

Comment 4: Line 46 - do these bacteria names need to be in Italics?

Response: Thanks very much. Italics have been used for bacterial names in the revised manuscript. (lines 54-56)

Comment 5: Line 58 - how big was the farm, how many animals did it have?

Response: Thanks very much. The commercial dairy farm where this study was conducted covers an area of approximately 530,000 square meters and houses approximately 4,200 head of cattle. We apologize for not including these details in our manuscript, as it is not a common practice in studies focusing on cattle in the Animals journal.

Comment 6: Line 67 - what is a colostrum baster?

Response: Thanks very much. We appreciate your attention to detail. We have made the necessary corrections in the revised manuscript to accurately describe the equipment as a "Colostrum Pasteurization Machine."  (line 74)

Comment 7: Line 71 - do not start a sentence with a number. Eight rather than 8.

Response: Thanks very much. Revised as you suggested. (line 79)

Comment 8: Line 78 - experiment rather than experimental

Response: Thanks very much. Revised as you suggested. (line 87)

Comment 9: Line 84 - exactly how much milk replacer powder was fed in g? It is unusual to give a ration of 1:6 for milk replacer.

Response: Thanks very much. Each meal of 5L of MR contains five-sevenths of milk powder. The ratio of milk powder to water is based on the recommendations of the MR manufacturing company.

Comment 10: Line 87 - the grammar of this sentence needs adjusting.

Response: Thanks very much. Revised as you suggested. (lines 94-96)

Comment 11: Line 108 - how were calves weighed?

Response: Thanks very much. All calves were weighed using electronic scales on an empty stomach before morning feeding. Additions have been made in the revised manuscript. (lines 116-117)

Comment 12: Line 118 - grammar of this sentence needs improving. Was this a commercial preparation of oral fluids? How were the fluids offered?

Response: Thanks very much. The grammar of the sentence has been corrected. And the company name of the oral rehydration solution has been added.  (line 126-130)

Comment 13: Line 157 - grammar of this sentence needs improving.

Response: Thanks very much. Revised as you suggested. (lines 167-171)

Comment 14: For section 2.8 and 2.9, much of the methodology in this section could be presented either in a flow chart of table to make it easier to follow.

Response: Thanks very much. After careful consideration, we have decided to maintain the current format of the methodology in these sections. While we acknowledge that a flow chart or table may provide a more visually appealing presentation, we believe that the textual description adequately conveys the necessary information. We would like to note that this approach is consistent with the style and format of similar studies published in our field. Additionally, the textual description allows for a more detailed and comprehensive explanation of the methodology, which may be beneficial for readers seeking a deeper understanding of the experimental procedures.

Comment 15: Line 193 - does not need a hyphen between soft-ware

Response: Thanks very much. Revised as you suggested. (line 203)

Comment 16: Table 2 - highlight the p-value for starter DMI as being P<0.1 for a trend. This is usually done with a # or similar mark.

Response: Thanks very much. Revised as you suggested. (table 2, table 9, line 219, 296)

Comment 17: Line 217 - calf not calve

Response: Thanks very much. Revised as you suggested. (line 228)

Comment 18: Figure 1 - the use of the lower case letter is confusing. Please change this to some other method of identifying differences. This also applies to the tables. This needs changing - what is the difference between a and ab and b?

Response: Thanks very much. We understand your concern regarding the potential confusion caused by the use of lowercase letters in our manuscript. However, we would like to highlight that the letter marking system is a widely accepted and commonly used method in statistical analysis to represent significant differences among multiple groups, as is the case in the article provided below where the letter marking method is used. The absence of the same lowercase letter between values indicates a significant difference (P < 0.05). Thus there is no significant difference between a and ab, a significant difference between a and b, and no significant difference between ab and b.

       Elwakeel, E.A.; Titgemeyer, E.C.; Cheng, Z.J.; Nour, A.M.; Nasser, M.E. In Vitro assessment of the nutritive value of expanded soybean meal for dairy cattle. Journal of animal science and biotechnology 2012, 3, 10, doi:10.1186/2049-1891-3-10.

Comment 19: Line 233 - does medium dose refer to the low dose of capsaicin?

Response: Thanks very much. Yes, the correct expression here should be low dose, which has been modified in the revised manuscript. (line 244)

Comment 20: Line 255 - does this mean there is no significant difference between the groups?

Response: Thanks very much. Yes, there were no significant differences in the results of alpha diversity analysis and PCoA plot for the fecal microbiota of each group.

Comment 21: Line 299-314 - some of this information should be moved up to the introduction section of the article as it is background to how Capsaicin has been used before.

Response: Thanks very much. Revised as you suggested. (lines 51-53, 311-326)

Comment 22: Line 321 - space missing  between diet. CR

Response: Thanks very much. Revised as you suggested. (line 332)

Comment 23: Line 326 - change the term lactating calves as mentioned above

Response: Thanks very much. Revised as you suggested. (line 337)

Comment 24: Line 390 - Does xin need a capital letter, and why is the reference repeated at the end of the sentence?

Response: Thanks very much. The repeated mention at the end of the sentence is due to my oversight. I have changed xin to Xin in the revised manuscript and deleted the repeated part at the end of the sentence.  (line 406)

Comment 25: Line 400 - Capital letter for The

Response: Thanks very much. Revised as you suggested. (line 415)

Comment 26: Line 412 -  re-phrase the 'in conclusion' here as this sentence is followed by the papers conclusion.

Response: Thanks very much. Revised as you suggested. (lines 430-433)

Thank you and all the reviewers for the kind comments.

Sincerely yours,

Minqiang Su